# Rapid HPLC-ESI-MS/MS Analysis of Neurotransmitters in the Brain Tissue of Alzheimer’s Disease Rats before and after Oral Administration of *Xanthoceras sorbifolia* Bunge

**DOI:** 10.3390/molecules23123111

**Published:** 2018-11-28

**Authors:** Zheng Sun, Qing Li, Kaishun Bi

**Affiliations:** National and Local Joint Engineering Laboratory for Key Technology of Chinese Material Medica Quality Control, School of Pharmacy, Shenyang Pharmaceutical University, 103 Wenhua Rd., Shenyang 110016, China; sunzheng024@163.com (Z.S.); liqing@syphu.edu.cn (Q.L.)

**Keywords:** neurotransmitters, *Xanthoceras sorbifolia* Bunge., HPLC–MS/MS, Alzheimer’s disease

## Abstract

In order to explore the potential therapeutic effect of *Xanthoceras sorbifolia* Bunge. against Alzheimer’s disease, an HPLC-MS/MS method has been developed and validated for simultaneous determination in rat brain of eight neurotransmitters, including dopamine, norepinephrine, 5-hydroxy-tryptamine, acetylcholine, l-tryptophan, γ-aminobutyric acid, glutamic acid and aspartic acid with a simple protein precipitation method for sample pre-treatment. The brain samples were separated on a polar functional group embedded column, then detected on a 4000 QTrap HPLC-MS/MS system equipped with a turbo ion spray source in positive ion and multiple reaction monitoring mode. The method was fully validated to be precise and accurate within the linearity range of the assay, and successfully applied to compare the neurotransmitters in the rat brain from four groups of normal, Alzheimer’s disease, and the oral administration group of *X. sorbifolia* extract and huperzine. The results indicated that brain levels of dopamine, norepinephrine and acetyl choline all decreased in the AD rats, while l-tryptophan showed an opposite trend. After administration of the *Xanthoceras sorbifolia* extract and huperzine, the level of acetyl choline and tryptophan returned to normal. Combination of the metabolic analysis, the results indicated that acetyl choline and l-tryptophan could be employed as therapy biomarkers for AD, and the results shown that the crude extract of the husks from *Xanthoceras sorbifolia* might ameliorate the impairment of learning and memory in the Alzheimer’s disease animal model with similar function of AchEI as huperzine. The established method would provide an innovative and effective way for the discovery of novel drug against Alzheimer’s disease, and stimulate a theoretical basis for the design and development of new drugs.

## 1. Introduction

Alzheimer’s disease (AD) is an age-related progressive neurodegenerative disease with prominent neuropathologic features of senile plaques, neurofibrillary tangles, neuroinflammation, synaptic and cell loss [1,2]. In recent decades, it is among the most prevalent forms of dementia affecting the aging population. Although intensive researches have been done, there is no cure and early preclinical diagnostic assays available for AD so far [3,4]. Neurotransmitters are basic signaling small molecules which are widely distributed in central neural system and body fluids of mammals [5,6]. Levels of neurotransmitters in bio-samples, e.g., brain tissue, cerebrospinal fluid, plasma and urine, are implicated among metabolic system, regulatory system and immunity system. Compared to normal people, the neurotransmitters level of AD patients typically changed remarkably [7,8]. In addition, many scientific studies have demonstrated the importance of metabolites analysis in body fluids of AD patients in relationship to the discovery of novel drug targets. Therefore, the key point of AD study is now being focused more and more on the changes of the neurotransmitter levels in bio-samples such as plasma and brain [4,9].

*Xanthoceras sorbifolia* Bunge. (X. sorbifolia), as a common shrub, is widely distributed in the northwestern and northeastern regions of China with a lifespan of more than 200 years [10]. It has been used as folk medicine for its excellent treatment on rheumatism and enuresis of children. In recent years, it has been reported that the neurotransmitters in *X. sorbifolia* showed multiple bioactivities, such as anti-inflammatory, anti-HIV, antitumor activities, and especially, the function of improving intelligence [11,12,13,14]. For example, xanthoceraside, a triterpenoid saponin isolated from the fruit husks of *X. sorbifolia*, was reported to exhibit protective effects on the spatial memory impairment and oxidative stress induced by intracerebroventricular (i.c.v.) injection of Aβ25–35 or Aβ1–42 in rats [15,16]. In order to find new candidates against AD, we have focused our interest on the effect of *X. sorbifolia* on AD model rats. With advantages of multi-target and multi-coordinated system, Chinese herbs are being widely used in various illness treatments, and positive effects have been achieved. However, problems, including incomplete functional mechanisms, have affected the application of traditional Chinese medicines (TCMs), so it will be important for the use of TCMs to combat AD to obtain pharmacodynamic evidence on the basis of neurotransmitter metabolism.

Nowadays, several analytical methods have been developed and applied for quantitative analysis of neurotransmitters, including radioenzymatic assays, capillary electrophoresis, gas chromatography, high performance liquid chromatography (HPLC) with ultraviolet detector (UV), fluorimetric or electrochemical detection and the combination of HPLC and mass spectrometry [17,18,19,20,21,22]. However, most of these methods are laborious, requiring pre-column or post-column derivatization and time-consuming sample preparations with long chromatographic separations. Recently, the high performance liquid chromatography tandem mass spectrometry (HPLC-MS/MS) methods have gained popularity in neuroscience research due to their high sensitivity, specificity and applicability for complex matrices such as tissues and body fluids [5,23]. When a triple quadrupole analyzer coupled with multiple reaction monitoring (MRM) was employed, neurotransmitters always exhibited higher signals, and the selectivity can be improved significantly.

Therefore, in our study, a rapid and sensitive method for the analysis of neurotransmitters in the brain of Alzheimer’s disease rats was developed, which was accordingly applied to successful quantification of eight neurotransmitters in the brain of the AD rats, *X. sorbifolia* extract orally administered rats, huperzine treated rats (positive control rats) and normal rats. Moreover, the impact of AD on neurotransmitter levels was investigated by examining brain samples from D-gal and Aβ25–35 (amyloid beta peptide25–35)-induced AD rats. Then, the altered neurotransmitters level in the presence of *X. sorbifolia* extract was evaluated. By a targeted metabolomics analysis, an innovative protocol combining simultaneous determination of eight neurotransmitters to identify differential neurotransmitters for early AD diagnosis, and screen out sensitive biomarkers for monitoring the therapeutic effects of the drug, has been constructively developed. The results obtained could provide a theoretical basis for the design and development of novel drugs. 

## 2. Results

### 2.1. HPLC-MS/MS Method Development and Optimization

#### 2.1.1. Optimization of Chromatographic Conditions

As most neurotransmitters and their metabolites are nitrogenous aliphatic compounds with highly charged cations, the retention behavior on C_18_ columns and the detection by mass spectrometry are typically unsatisfactory. As a result, an Inertsil ODS-EP column and gradient elution were used to improve the retention and separation. The column used in this method was chosen since it gave better retention behavior of the compounds with high polarity in the high aqueous mobile phase in comparison with other reversed phase columns. The column contains a polar functional group embedded between the silica surface and C_18_ group, so it offers excellent selectivity as hydrogen bonding interactions are provided by the polar-embedded functional groups. Methanol was chosen as the organic phase as it offered higher mass spectrometric response and lower background noise compared with acetonitrile.

Furthermore, the addition of acid to the mobile phase brought obvious improvement of the peak shape and sensitivity for most of the compounds. As the mobile phase additive in reversed phase liquid chromatography, heptafluorobutyric acid (HFBA) can prolong retention via forming ion pairs with cationic compounds, and improve peak shape by suppressing the interaction between nitrogenous compounds and free silanol groups of the column as well. However, the ion response of 5-hydroxytryptamine and dopamine were significantly decreased by increasing the concentration of the acid in mobile phase, which could also result in a poor peak shape for tryptophan. Consequently, 0.01% HFBA was adopted, which possessed the optimal signal intensities and symmetrical peak shape for all the analytes in a relatively short run time.

#### 2.1.2. Optimization of Mass Spectrometric Conditions

In the experiment, the mass spectrometry (MS) operation parameters were optimized for determination of the eight analytes in brain samples. The positive ion mode and electrospray ionization (ESI) source were applied based on the characteristics and the optimization results of the compounds. The optimal conditions for the triple-quadrupole mass analyzer were determined by direct infusion of the standard solutions into the mass spectrometer (see Appendix A). Precursor and product ions with excellent ion performance were chosen for quantitation and unequivocal identification (Table 1). The critical parameters such as Gas1, Gas2, curtain gas, declustering potential (DP), collision energy (CE) and cell exit potential (CXP) were optimized in automatic runs under evaluation of the ionic sensitivity. Other parameters were adopted for recommended values of the instrument. Typical MRM chromatograms of the eight compounds in rat brain samples of the four groups were shown in Figure 1.

### 2.2. Morris Water Maze Testing

After two weeks of injection with Aβ25–35 (amyloid beta peptide25–35), the Morris water maze test was performed for 5 days. Rats were trained twice a day for four consecutive days with an inter-trial interval of 3 h, then allowed to escape by swimming to the platform, and the escape latency was recorded for 90 s. After 90 s, if the rats failed to locate the platform, they were then placed on the platform for 20 s. On the fifth day, the rats were given a probe test. In this test, the platform was removed, and each rat was allowed to explore the pool for 90 s. The escape latency (time spent in the target quadrant), distance swum, and swimming speed were measured using a computer system with a video camera. The results demonstrated that the rats in the AD modeling group took longer time to reach the platform than other three groups at the fourth, fifth, sixth and seventh trail (*p* < 0.05), which verified the success of the AD modeling (see Appendix A). 

### 2.3. Method Validation

#### 2.3.1. Linearity and Lower Limit of Quantitation

The linear range and regression equations for the quantification of the 8 analytes were presented in Table 2. The assay was found to be linear for all the analytes under investigation over the concentration range. The Lower Limit of Quantitation (LLOQ) of this method was proved to be the lowest concentration on the calibration curve with a signal-to-noise ratio greater than 10, which was qualified to perform the determination. In addition, the precision and accuracy of LLOQ (Table 2) were well within acceptance criteria (20%).

#### 2.3.2. Precision and Accuracy

Intra- and inter-day precision and accuracy were determined by six replicate analyses of Quality Control (QC) samples at three concentration levels within the same day and on three consecutive days using freshly prepared calibration curves. Results listed in Table 3 confirm the acceptable values of the proposed method.

#### 2.3.3. Recovery and Matrix Effect

Recoveries of the eight neurotransmitters were all above 60% at different concentration levels, and the results are summarized in Table 3. The IS recovery was 78.32%. The results showed that the recoveries of all the analytes were consistent, precise and reproducible at different concentrations.

Matrix effects (MF) of each analyte, which was evaluated at three QC levels were listed in Table 3. The obtained result showed that the RSDs of the IS-normalized MF were all no greater than 11% (Table 3), indicating that no significant matrix effect for the analytes and IS were observed. Thus, no co-eluting substance influenced the ionization of the analytes. 

#### 2.3.4. Stability

It was observed that the eight brain analytes were stable after being placed at room temperature with RSD all less than 8.3% within 4 h and the reconstituted extract analytes were stable at 4 °C in the auto-sampler for 4 h with all the RSDs less than 8.8%. Moreover, all the analytes were stable when the QC samples were stored at −80 °C for 1 month with all the REs less than 7.9%. The eight analytes were also stable through three freeze-thaw cycles with all the REs less than 10.8% and all the RSDs less than 9.5%. The results indicated that the three analytes in brain samples were stable under different storage conditions.

## 3. Discussion

The proposed method was then applied for determination of the eight neurotransmitters in rat brain samples from four groups. Mean concentrations of these endogenous compounds were illustrated in Table 4. It was shown that the levels of norepinephrine, acetylcholine and glutamic acid in AD rats (Group M) were significantly lower than that in the normal group (*p* < 0.05), whereas the content of tryptophan in AD rats (Group M) appeared to be higher than the normal group (*p* < 0.05). However, no remarkable differences of 5-hydroxytryptamine, γ-aminobutyric acid, glutamic acid and aspartic acid between the two groups were observed. Relative reports of neurotransmitters in brain tissues of AD patients revealed decreased concentrations of acetylcholine, norepinephrine and glutamic acid [9,24]. Our results were basically in accordance with the previous literatures. Since tryptophan metabolic network was extremely complicated, reports of the changes of tryptophan level were highly varied, such as reduced in AD, reduced in mild cognitive impairment, unchanged in AD and raised in AD were all observed in different researches [25,26,27]. In this study, it had been found that the increased brain tryptophan level in the AD model group. Because of the successful induction of AD rats by D-gal and Aβ25-35 verified by behavioral testing and previous papers [8,9], this phenomenon might be attributed to the difference of certain enzyme activities between rats with AD and normal controls. In our follow-up experiment, we will focus more attention on certain enzyme activities and make an in-depth investigation the tryptophan metabolic network of AD.

Figure 2 showed the significant varied neurotransmitters in the four animal groups. Compared with AD rats (Group M), the content of acetylcholine increased, whereas the content of norepinephrine, glutamic acid and tryptophan decreased in the *X. sorbifolia* medication group (Group X), and the variation trend of the four analytes was similar to that of the huperzine medication rats (Group H). Huperzine, an alkaloid, is widely used to alleviate the symptoms of AD as an acetylcholinesterase inhibitor [28,29]. In the course of AD disease, one of the most important pathological factors for dementia is the deterioration of the cholinergic neuron, and the major mode of deterioration is cell apoptosis. The principle neurotransmitter of the forebrain cholinergic system in basal ganglia is acetylcholine. Therefore, at present, the most widely studied anti-AD drugs are acetylcholinesterase inhibitors (AchEIs) [30,31,32]. Meanwhile, the experimental results showed that the extract of *X. sorbifolia* might exhibit a similar AchEI function as huperzine. From the analysis of brain neurotransmitters in *X. sorbifolia* and huperzine medication rats, the level of l-tryptophan in both the *X. sorbifolia* (Group X) and huperzine medication rats (Group H) dropped. According to the previous literature, different neurotransmitters, downstream products of l-tryptophan, play different roles, whereby some exhibit neuroprotective activity, whereas others exhibit neurotoxic activity. For norepinephrine and glutamic acid, *X. sorbifolia* and huperzine might affect upstream or downstream products in their metabolic pathways. Thus for evaluating the therapeutic efficacy of anti-AD drug, the metabolic pathway of l-tryptophan and acetylcholine may the key indicators. Further studies on neurotransmitters and their metabolic pathways might be necessary in investigating the mechanism of AD and as well as its treatment. In conclusion, both l-tryptophan and acetylcholine may have similar characteristics: a significant change level in accordance with AD progression and a regulated tendency according to the treatment until it becomes equal to that in normal rats. Thus, we concluded that l-tryptophan and acetylcholine could serve as potential target biomarkers for monitoring medical treatments and predicting AD remission.

## 4. Materials and Methods

### 4.1. Chemicals and Standards

Dopamine (DA), norepinephrine hydrochloride (NE), 5-hydroxytryptamine (5-HT), acetyl- choline (Ach), l-tryptophan (Trp), γ-aminobutyric acid (GABA), glutamic acid (Glu) and aspartic acid (Asp) were purchased from Sigma-Aldrich (St. Louis, MO, USA). Isoproterenol hydrochloride (internal standard, IS) was supplied by the National Institute for Food and Drug Control (Beijing, China). Heptafluorobutyric acid (HFBA) was obtained from Sigma-Aldrich. The purity of these reference standards were all more than 98%. Methanol and acetonitrile (HPLC grade) were purchased from Fisher Scientific (Fair Lawn, NJ, USA). Distilled water prepared with demineralized water was used throughout the study. Other reagent and solvent of analytical grade, were provided by the department of pharmaceutics, Shenyang Pharmaceutical University (Shenyang, China).

### 4.2. Animals

Forty male Sprague-Dawley rats (weight 220–250 g) were obtained from the Experimental Animal Center of Shenyang Pharmaceutical University and bred with unlimited access to food and water in an air-conditioned animal center at a temperature of 22 ± 2 °C and a relative humidity of 50 ± 10%, with a natural light-dark cycle for a week to adapt to the environment. All animals were starved for 12 h before the experiment but with access to water throughout the study. The animal study was carried out following the Guideline of Animal Experimentation of Shenyang Pharmaceutical University, and the protocol was approved by the Animal Ethics Committee of the university.

The animals were divided randomly and equally into four groups: the normal group (Group N), the model group (Group M), the *X. sorbifolia* extract administration group (Group X) and the huperzine administration group (Group H). The AD modeling of the rats in Group M, X and H were induced by D-gal and Aβ25–35, according to the preciously reports [9]. Briefly, the rats in Group M, X and H were given intra-peritoneal injection of D-gal (50 mg·kg^−1^·day^−1^) for 42 days continuously and the rats in Group N were given the same volume of saline. At the beginning of the second week, Group X and H were orally administered *X. sorbifolia* (950 mg·kg^−1^) and huperzine (0.027 mg·kg^−1^) everyday respectively, and Groups N and M were orally administered the same volume of water everyday. The administrated process lasted for 28 days. Then, at the end of the last administration, the rats in Group M, X and H were individually injected with 4 μg Aβ25–35 (1 μg·μL^−1^, dissolved in sterilizing saline) into each bilateral hippocampus at the antero-posterior, −3.5, mediolateral, +2.0, dorsoventral, 3.0 mm co-ordinates according to the stereotaxic atlas by s brain stereotaxic apparatus for one time, and the rats in Group N were injected the same volume of sterile saline. Subsequently, after 14 days of injection with Aβ25–35, all the rats were subjected to a Morris water maze test for 5 days. Twenty-four hours after Morris water maze testing, the rats were sacrificed by decapitation without anesthesia, then brain samples were rapidly removed above an ice bath, and collected for immediate histopathological examination. The samples for neurotransmitter determination were frozen and stored in liquid nitrogen until extraction.

### 4.3. Instrumentation and chromatographic conditions

An LC-20A Prominence™ UFLC XR ultra high performance liquid chromatography system equipped with a binary pump, a degasser, an autosampler and a thermostatted column oven (Shimadzu, Kyoto, Japan) was employed in the study. Chromatographic separation was performed on an Inertsil ODS-EP column (150 mm × 4.6 mm, 5 μm, GL Sciences, Tokyo,.Japan) protected by a high pressure column pre-filter (2 μm, Shimadzu) at 30 °C. Chromatographic separation was achieved by gradient elution using a mobile phase consisted of 0.01% HFBA in water (A) and 0.01% HFBA in methanol (B). The LC gradient program was as follows: 10% → 20% B at 0.01–3.00 min; 20% → 85% B at 3.01–7.00 min; 85% → 90% B at 7.01–8.00 min; 10% B at 8.01–12.00 min. Efficient and symmetrical peaks were obtained using a tee joint (split ratio of load to waste was 1: 1) at a flow rate of 1.0 mL·min^−1^ with a sample injection volume of 5 μL. The detection of the analytes was performed by a 4000 QTrap triple quadrupole—linear ion trap mass spectrometer (Sciex, Foster City, CA, USA) equipped with a turbo ion spray source in MRM mode. The positive electrospray ionization (ESI^+^) was used. The ion spray voltage was set at 5500 V and the temperature was maintained at 500 °C. Gas 1 and Gas 2 were set at 50 psi, while the curtain gas was set at 20 psi. Quantitative parameters are listed in Table 1. Instrumental acquisition and data analysis were operated by the Analyst software (version 1.6, Sciex, Redwood, CA, USA).

### 4.4. Preparation of the Xanthoceras Sorbifolia Bunge. Extract and Huperzine Administration Solution

Dried husks (200 g) of *X. sorbifolia* were powdered and further extracted three times with 70% ethanol under reflux for 2 h. The mixture was filtered and the filtrates were combined and concentrated under reduced pressure to a density of 3.33 g crude drug per milliliter. The decoction was stored in the refrigerator at 4 °C for use. Huperzine, used as positive drug, has been found through multiple studies to be effective as a medicine for helping people with neurological conditions such as Alzheimer’s disease, were prepared in water at the concentration of 0.027 mg·ml^−1^.

### 4.5. Preparation of Standard Solutions and Calibration Curve Standards

The stock standard solutions of NE, DA, 5-HT, Ach, Try, GABA, Glu and Asp were prepared in 80% methanol/ water (*v*/*v*) at the concentrations of 4 μg·mL^−1^, 3 μg·mL^−1^, 2 μg mL^−1^, 3 μg·mL^−1^, 0.05 μg·mL^−1^, 0.05 μg·mL^−1^, 3 μg·mL^−1^ and 3 μg·mL^−1^, respectively. The stock standard solution of IS was diluted to a concentration of 0.375 μg·mL^−1^ with methanol as working solution. A series of mix standard solutions of desired concentrations were prepared by suitable dilution of the stock solutions. All the solutions were then stored at −20 °C.

The calibration standards were prepared by spiking brain homogenates from normal rats (n = 10) with the corresponding mixed standard solutions and IS. The solutions were then further diluted with methanol to achieve seven calibration standard solutions at the concentration range of 40.00–4000 ng·mL^−1^ for NE and Glu; 30.00–3000 ng·mL^−1^ for DA, Ach, GABA, and Asp; 20.00–2000 ng·mL^−1^ for 5-HT; 0.5000–50.00 ng·mL^−1^ for Glu and Try. Quality control (QC) samples were prepared in the same way (60.00, 375.0, 2400 ng·mL^−1^ for DA, Ach, GABA and Asp; 80.0, 500.0, 3200 ng·mL^−1^ for NE; 40.00, 250.0, 1600 ng·mL^−1^ for 5-HT; 1.000, 6.250, 40.00 ng·mL^−1^ for Glu and Trp).

### 4.6. Sample Preparation

The brain samples were collected after Morris water maze testing. All the samples (samples, standards and QC samples) were prepared by adding IS solution in order to compensate for any possible bias in accuracy originating from the sample preparation. The brain tissue (about 0.1 g) from each animal was weighted precisely and homogenized in ice bath with a 10-fold (*w*/*v*) volume of methanol and were then centrifuged at 13000 g for 5 min at 4 °C. 200 μL of supernatant was spiked with 25 μL of IS and 25 μL of methanol followed by vortexing for 3 min, then the mixtures was centrifuged at 13000 g for 5 min at 4 °C, and was then directly injected for analysis.

### 4.7. Method Validation

The method was completely validated according to the currently accepted US FDA Bioanalytical Method Validation Guidance and other related guidelines [33,34], including the characteristics of linearity, lower limit of quantification (LLOQ), precision, accuracy, recovery, matrix effect and stability. The calibration standards and QC samples were prepared by adding the dilution of the stock analyte solutions to the blank brain samples on every validation day. Concentrations of analytes in the QC samples were calculated by using calibration curves prepared on the same day. The samples were extracted as described in the section of sample preparation and analyzed by HPLC-MS/MS.

The linearity of the eight analytes was demonstrated with a total of seven calibration standards in the concentration range. Because of the presence of endogenous analytes in the brain, blank values of the eight neurotransmitters were subtracted from each calibration point. Calibration curves were then evaluated by least-square linear regression of the 8 analytes-to-IS peak area ratios (y) versus the normalized standard concentration (x) with a weighed (1/x^2^) factor at three different analytical batches. The accuracy and precision of intra-day and inter-day determination were carried out in six replicates at three QC levels (low, medium and high) within the same day and on three consecutive days.

The recovery was calculated of QC samples on six duplicates as Ae/As, where Ae is the response of extracted analytes or IS subtracted those of blank samples and As is the response of post-extracted blank plasma samples spiked with analytes subtracted those of blank samples. The matrix effect was evaluated as follows: for each of the eight analytes and the IS, the matrix factor (MF) was calculated by calculating the ratio of the peak area in the presence of matrix (measured by analyzing blank matrix spiked after extraction with analytes at three concentration levels subtracted those of blank samples), to the peak area in absence of matrix (pure standard solutions of the analytes at three concentration levels). The IS-normalized MF was calculated by dividing the MF of the analyte by the MF of the IS.

### 4.8. Statistical Analysis

Neurotransmitters concentrations were obtained from calibration curves and expressed as mean ± SD. The results were presented as mean ± SD and analyzed statistically with Student’s t-test and the nonparametric Mann-Whitney test using SPSS 19.0 software for Windows (SPSS Inc., Chicago, IL, USA). The threshold of significance was set at *p* < 0.05.

## 5. Conclusions

In the study, an HPLC-MS/MS method for simultaneous determination of eight neurotransmitters in the brain of Alzheimer’s disease rats was developed and further validated and the differences of their levels among four groups of normal, Alzheimer’s disease, and the oral administration groups of *X. sorbifolia* extract and huperzine were specifically analyzed. Results showed that decreased dopamine, norepinephrine and acetylcholine levels and increased tryptophan levels were associated with progression of AD. In addition, no remarkable differences of 5-hydroxytryptamine, γ-aminobutyric acid, glutamic acid and aspartic acid were observed between normal and pathological rats. After administration of the *X. sorbifolia* extract and huperzine, the levels of acetylcholine and l-tryptophan returned to normal. However, the level of norepinephrine and glutamic acid showed an opposite trend, which also illustrated the effect of *X. sorbifolia* and huperzine through the metabolite pathways. The developed method was rapid, sensitive and only small volume of biosample was required, and concentrations determined by the current method are in the same order of magnitude as in the literature. The established method was employed in analyzing neurotransmitters in brain tissues of AD model and *X. sorbifolia* orally administered animals, which fully proved that the crude extract of the husks from *X. sorbifolia* could significantly ameliorate the impairment in learning and memory in AD rats. Our results also demonstrated the application value of brain neurotransmitters in medical treatment of AD. Moreover, the method is expected to be especially useful for the discovery of novel drugs against AD.

## Figures and Tables

**Figure 1 molecules-23-03111-f001:**
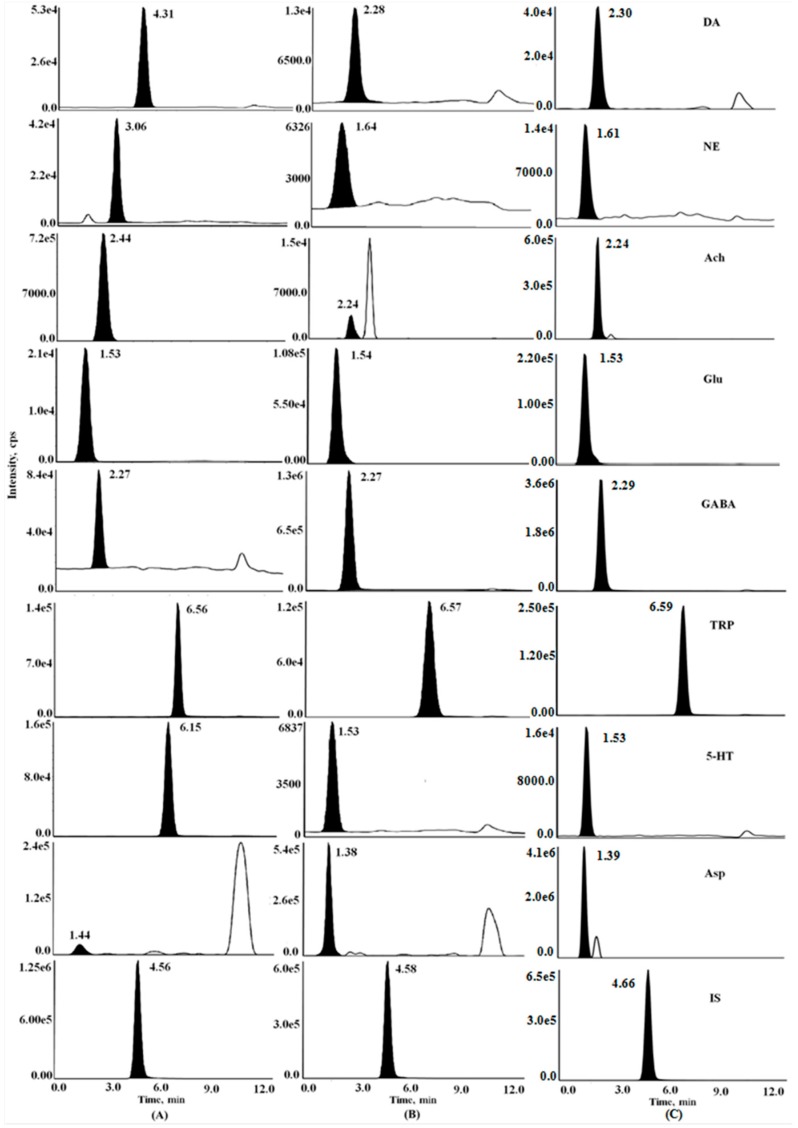
Typical multiple reaction monitoring chromatograms of eight standards and internal standard (**A**), the brain samples of the AD rats (**B**) and a blank sample spiked analytes and IS (**C**). DA, Dopamine; NE, norepinephrine; 5-HT, 5-hydroxytryptamine; Ach, acetyl choline; Trp, l-tryptophan; GABA, γ-aminobutyric acid; Glu, glutamic acid; Asp, aspartic acid.

**Figure 2 molecules-23-03111-f002:**
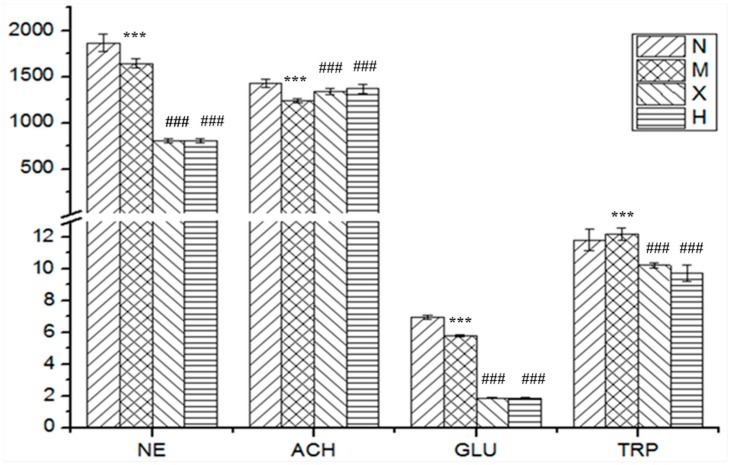
Mean concentration of four differential analytes (*p* < 0.05, analyzed by Student’s *t*-test and the nonparametric Mann-Whitney test) in rat brain in different groups (N: normal group; M: model group; X: *X. sorbifolia* group; H: huperzine administration group).NE, norepinephrine; Ach, acetyl- choline; Trp, l-tryptophan; Glu, glutamic acid. *** *p* < 0.001 compared with Group N, ^###^
*p* < 0.001 compared with Group M.

**Table 1 molecules-23-03111-t001:** List of multiple reactions monitoring parameters, declustering potential (DP), entrance potential (EP), collision energy (CE) and cell exit potential (CXP) for each analyte and isoproterenol hydrochloride (IS).

Analyte	Q1 Mass (*m*/*z*)	Q3 Mass (*m*/*z*)	DP (V)	EP (V)	CE (V)	CXP (V)
Dopamine	154.1	137.2	37	10	13	8
Norepinephrine	170.1	152.2	33	5	11	9
Acetylcholine	146.2	87.2	52	3	20	4
5-Hydroxytryptamine	177.1	160.2	43	10	14	10
Glutamic acid	148.2	84.1	41	10	24	15
γ-Aminobutyric acid	104.2	87.1	26	14	15	16
l-Tryptophan	205.2	188.0	40	4	14	12
Aspartic acid	134.0	74.0	38	9	20	13

**Table 2 molecules-23-03111-t002:** The linearity data and lower limit of quantification (LLOQ) with precision and accuracy for analytes in brain using the proposed method.

Analyte	Concentration Range (ng mg^−1^)	Regression Equation (r) (10^−4^)	LLOQ	
			Accuracy (RE%)	Precision (RSD%)
Dopamine	30.00–3000	y = 6.247x − 170.1 (0.9949)	8.4	5.0
Norepinephrine	40.00–4000	y = 1.231x − 6.498(0.9950)	−6.2	3.3
Acetylcholine	30.00–3000	y = 6.363x − 1.825(0.9926)	3.9	1.8
5-Hydroxytryptamine	20.00–2000	y = 12.42x − 94.77 (0.9949)	−11.3	13.5
Glutamic acid	0.5000–50.00	y = 259.3x − 18.57 (0.9914)	4.9	7.2
γ-Aminobutyric acid	30.00–3000	y = 136.9x − 1177 (0.9920)	7.8	4.8
l-Tryptophan	0.5000–50.00	y = 983.3x − 106.3 (0.9912)	−2.2	6.6
Aspartic acid	30.00–3000	y = 8.054x − 149.6 (0.9951)	4.8	10.2

RE, relative error; RSD, relative standard deviation.

**Table 3 molecules-23-03111-t003:** Summary of accuracy, precision, recovery and matrix effect in rat brain (n = 6).

Analytes	Conc (ng mg^−1^)	Intra-Day RSD%	Inter-Day RSD%	Accuracy RE%	Recovery (Mean ± SD%)	IS-Normalized MF (RSD%)
Dopamine	60	5.9	3.4	5.1	87.6 ± 4.7	5.4
	375	5.7	9.2	0.3	82.0 ± 4.9	7.6
	2400	10.1	6.4	7.0	79.45 ± 1.24	1.0
Norepinephrine	80	11.5	8.4	11.7	82.3 ± 6.8	8.7
	500	9.2	3.8	−4.3	66.94 ± 7.40	9.2
	3200	2.2	7.2	−2.8	71.97 ± 0.78	7.2
Acetyl choline	60	8.7	11.1	−10.1	70.62 ± 2.61	3.3
	375	11.4	5.9	9.7	73.06 ± 4.92	3.7
	2400	10.0	0.9	11.6	65.25±4.09	6.5
5-Hydroxytryptamine	40	10.4	8.8	−10.4	83.3 ± 5.3	9.7
	250	9.6	1.7	7.1	85.2 ± 7.2	0.4
	1600	4.6	8.4	−5.1	77.12 ± 2.27	10.1
Glutamic acid	1	11.4	5.6	8.4	82.8 ± 7.6	5.9
	6.25	2.9	11.0	6.5	67.00 ± 4.87	6.9
	40	5.5	7.1	3.7	79.92 ± 0.77	4.5
γ-Aminobutyric acid	60	4.5	3.2	−11.5	84.1 ± 0.9	7.9
	375	1.6	6.6	−10.3	69.53 ± 5.5	6.3
	2400	3.7	9.3	5.7	73.44 ± 8.83	10.7
l-Tryptophan	1	3.8	5.4	−5.1	81.2 ± 6.7	3.6
	6.25	6.0	7.9	9.6	81.1 ± 6.4	10.5
	40	4.4	1.8	2.0	67.70 ± 2.64	5.0
Aspartic acid	60	9.0	11.1	4.8	77.97 ± 1.40	9.9
	375	6.1	7.7	−11.9	85.8 ± 7.3	1.4
	2400	7.1	6.7	0.9	72.90 ± 7.62	0.8

RE, relative error; RSD, relative standard deviation.

**Table 4 molecules-23-03111-t004:** Mean concentration (ng·mg^−1^) of the analytes in rat brain in the four experimental groups (mean ± SD, n = 10).

Analytes	Group N	Group M	Group X	Group H
Dopamine	1132 ± 57	1157 ± 60	1111 ± 67	1084 ± 69
Norepinephrine	1868 ± 95	1601 ± 51 ***	808 ± 20 ^###^	807 ± 21 ^###^
Acetylcholine	1431 ± 47	1195 ± 21 ***	1340 ± 31 ^###^	1371 ± 49 ^###^
5-Hydroxytryptamine	305.0 ± 61.1	268.5 ± 54.9	283.8 ± 78.7	248.3 ± 42.3
Glutamic acid	6.977 ± 0.137	5.370 ± 0.068 ***	1.866 ± 0.045 ^###^	1.829 ± 0.060 ^###^
γ-Aminobutyric acid	618.2 ± 163.6	644.6 ± 128.5	494.4 ± 54.2	514.7 ± 70.1
l-Tryptophan	11.83 ± 0.67	13.56 ± 0.38 ***	10.23 ± 0.18 ^###^	9.73 ± 0.50 ^###^
Aspartic acid	533.2 ± 47.7	577.6 ± 65.7	635.3 ± 84.2	677.5 ± 83.2

*** *p* < 0.001 compared with Group N, ^###^
*p* < 0.001 compared with Group M. (N: normal group; M: model group; X: X. sorbifolia group; H: huperzine administration group).

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
