# Peer review of "Rapid HPLC-ESI-MS/MS Analysis of Neurotransmitters in the Brain Tissue of Alzheimer’s Disease Rats before and after Oral Administration of Xanthoceras sorbifolia Bunge"

_molecules, 2018, doi:10.3390/molecules23123111_

Round 1

Reviewer 1 Report

The present manuscript focuses on studying the effect of X. sorbifolia administration in brain levels of 8 neurotransmitters in a rat model of Alzheimer’s disease. For this purpose, authors optimized and validated a LC/MS-based approach, which was subsequently applied to brain samples. Although the topic is of interest, some modifications must be accomplished before publication.

·    Lines 81-82. Authors say that methanol was used as organic solvent, but acetonitrile is mentioned in section 4.3

·    Lines 98-99. Two MRM transitions should be monitored for each compound (quantifier and qualifier)

·    Discussion. A more detailed discussion of results must be performed by comparing with previously published studies. In this sense, a major limitation of the current work is that brain tissues are investigated as a whole organ. Authors have to mention that NT alterations in Alzheimer’s disease highly depends on the brain region investigated, being hippocampus and cortex the most affected regions. For this purpose, include as references some papers describing the measurement of NTs (both targeted and metabolomics approaches) in various brain regions.

·    Line 202. Change this title to “Chemicals and standards”, or similar

·    Section 4.2. Animal handling must be explained in more detail. D-Gal and Aβ were daily administered? What about X. sorbifolia and huperzine? When were the rats sacrificed?

·    Line 237. According to the manuscript title, an UHPLC-MS/MS method has been developed. However, a 5 μm LC column is mentioned in section 4.3. Please, clarify.

·    Lines 275-276. Which procedure was employed for brain homogenization?

·    Revise the written English along the entire manuscript (typographical errors, verb conjugations)

Author Response

Dear reviewer:

Thank you so much for reviewing and giving us the chance to revise our manuscript # molecules-388362 entitled “Rapid analysis of neurotransmitters in the brain tissue of Alzheimer's disease rats before and after oral administration of Xanthoceras sorbifolia Bunge by UHPLC-ESI-MS/MS ". We greatly appreciate the efforts for handling and reviewing our manuscript as well as the valuable comments. Respective revisions that provide significant help for the manuscript have been made. Please find our response to the reviewer point by point below and the relative changes made in the revised manuscript. 

Lines 81-82. Authors say that methanol was used as organic solvent, but acetonitrile is mentioned in section 4.3

Reply: We are sorry about the mistake. In this study, we use methanol as organic solvent, and we had corrected the error in section 4.3.

Lines 98-99. Two MRM transitions should be monitored for each compound (quantifier and qualifier)

Reply: As suggested by the reviewer, we should choose two ion pairs for the MRM determination. But for some analytes (MW about 100), only one stable daughter ion can be formed, so it is hard to monitored two MRM transition on all analytes. As a matter of fact, the adopted ion pairs for quantification in our study were all the ion pairs with the highest mass spectrometric response and all corresponded to the mass spectrometric fragmentation law of the fourteen analytes. Meanwhile, the eight analytes were authenticated from the comparison of mass spectrometric characteristics of its standards. Therefore, one ion pairs used for quantification is acceptable and convenient.

Discussion. A more detailed discussion of results must be performed by comparing with previously published studies. In this sense, a major limitation of the current work is that brain tissues are investigated as a whole organ. Authors have to mention that NT alterations in Alzheimer’s disease highly depends on the brain region investigated, being hippocampus and cortex the most affected regions. For this purpose, include as references some papers describing the measurement of NTs (both targeted and metabolomics approaches) in various brain regions.

Reply: According to the reviewer’s advice, we had rewritten the “Discussion” part. In this study, we aim to evaluate the effect of Xanthoceras sorbifolia Bunge by analysis of varied neurotransmitters level in brain tissue. And we do now to detect the neurotransmitters in different brain region to identify the differential neurotransmitters of AD and to further evaluate the whole regulative effect of Xanthoceras sorbifolia Bunge.

Line 202. Change this title to “Chemicals and standards”, or similar

Reply: Thanks a lot for the advice. We had changed the title as you suggest.

Section 4.2. Animal handling must be explained in more detail. D-Gal and Aβ were daily administered? What about X. sorbifolia and huperzine? When were the rats sacrificed?

Reply: Thanks for the advice. As suggested by the reviewer, we had rewritten the section 4.2 and explained the animal modeling process. The rats were given intra-peritoneal injection of D-gal (50 mg·kg-1·day-1) for 42 days continuously and then were oral administrated with X. sorbifolia (950 mg/kg) and huperzine (0.027 mg·kg-1) for 28 days continuously, Then, at the end of the last administration, the rats were injected with 4 μg Aβ25-35 (1 μg·μl-1, dissolve in sterilizing saline) for one time. After 14 days of injection with Aβ25-35, all the rats experienced the Morris water maze test for 5 days Twenty-four hours after Morris water maze testing, the rats had been sacrificed.   

Line 237. According to the manuscript title, an UHPLC-MS/MS method has been developed. However, a 5 μm LC column is mentioned in section 4.3. Please, clarify.

Reply: Thank the reviewer for the comments. We are sorry for our misunderstanding. In this study, we use an Inertsil ODS-EP column (150 mm × 4.6 mm, 5 μm) for neurotransmitters separation. It was an HPLC-MS/MS method, and we had corrected all the inaccurate expression in the manuscript.

Lines 275-276. Which procedure was employed for brain homogenization?

Reply: A homogenizer had been employed for brain homogenization, and each brain tissue was first weighted precisely and then homogenized in ice bath. In the process of homogenization, the brain tissue had been well-dispersed in a 10-fold (w/v) volume of methanol and been centrifuged at 13000 g for 5 min at 4 °C.

Revise the written English along the entire manuscript (typographical errors, verb conjugations)

Reply: Thank the reviewer for the comments. As suggested by the reviewer, we have revised the manuscript and tried to avoid grammar or syntax error.

Thanks again for the helpful advices. All the revisions provided significant help to make the manuscript more accurate and easier to understand. We hope that the revisions we made are satisfactory and the revised version of the manuscript will be acceptable for publication in Molecules. If you have any queries, please do not hesitate to contact me.

Wish you all the best!

Yours sincerely,

Kaishun Bi, Ph.D.

Professor

Shenyang Pharmaceutical University

Reviewer 2 Report

Sun et al. have developed a method to measure 8 neurotransmitter molecules and applied these to brains from a rat model of chemically-induced Alzheimer's disease. My major issue with this manuscript is that it seems this group have developed a method of almost identical use (with only the addition of aspartate in the current manuscript) to a previous manuscript which actually showed an improved method in terms of recovery etc. (He et al. J Mass Spec 2013;48:969-978). It is therefore not understood why the authors have 'redeveloped' an already successful method and then constructed a manuscript that is focussed on method development/validation with only a small aspect on the results of the targeted study. In addition to the need for major English editing, I have a number of other major issues with the manuscript:

- The authors claim they are measuring molecules such as norepinephrine hydrochloride, aspartic acid, glutamic acid etc. where in fact this is not technically true. Although the standard chemicals may be purchased in this format, the LC-MS/MS analysis is only able to quantitate on the levels of the salt (aspartate, glutamate etc.).

- There are no data presented in the abstract at all, this is not informative to the readers.

- A statement is made on the 'excellent treatment' using X sorbifolia but no references are provided.

- Abbreviations are used without definition (CE, HPLC, UV).

- The technical language used is, at best, poor (e.g. much higher signal).

- The end of the introduction states 'two groups', where as later it is discussed as four.

- Results should not be provided in the discussions section, these should be in the results section.

- Figure 2 shows 4 NTs, the figure legend says there are 8.

- The discussion is overall extremely weak and confusing and comments made are entirely speculative and don't warrant inclusion without sufficient evidence to suggest that the predictions 'could' be true. Solely measuring values does not provide the evidence in this context.

- Huperzine is mentioned as a treatment to one group - at no point is it explained what huperzine is, and its preparation details are poor - 0.027mg/kg is a dose, not a stock concentration (unless it is 0.027mg per 1 kg of solvent which I presume is not the case).

- The authors claim that only a small volume of biosample is required - but at no point do they mention how much brain tissue was used (an average or range, for example).

Author Response

Dear reviewer:

Thank you so much for reviewing and giving us the chance to revise our manuscript # molecules-388362 entitled “Rapid analysis of neurotransmitters in the brain tissue of Alzheimer's disease rats before and after oral administration of Xanthoceras sorbifolia Bunge by UHPLC-ESI-MS/MS ". We greatly appreciate the efforts for handling and reviewing our manuscript as well as the valuable comments. Respective revisions that provide significant help for the manuscript have been made. Please find our response to the reviewer point by point below and the relative changes made in the revised manuscript.

The authors claim they are measuring molecules such as norepinephrine hydrochloride, aspartic acid, glutamic acid etc. where in fact this is not technically true. Although the standard chemicals may be purchased in this format, the LC-MS/MS analysis is only able to quantitate on the levels of the salt (aspartate, glutamate etc.).

Reply: Thank the reviewer for the comments. As suggested by the reviewer, we quantitated the molecule and purchased the salt format. In the revised paper, we had corrected the error.

There are no data presented in the abstract at all, this is not informative to the readers.

Reply: Thanks a lot for the advice, we had rewritten the “Abstract” part.

A statement is made on the 'excellent treatment' using X sorbifolia but no references are provided.

Reply: Thank a lot for the advice. Xanthoceras sorbifolia Bunge. has been used for more than 200 years as folk medicine with anti-inflammatory, anti-HIV, antitumor activities, and especially, the function of improving intelligence. And Xanthoceraside had been reported to exhibit protective effects on the spatial memory impairment and oxidative stress. The related references had been added in the revised manuscript now.

Abbreviations are used without definition (CE, HPLC, UV).

Reply: Thank you for the suggestions, these abbreviations had been replaced by the full spelled.

The technical language used is, at best, poor (e.g. much higher signal).

Reply: We are sorry about the writing mistake, we had revised the manuscript and tried to avoid grammar or syntax error.

The end of the introduction states 'two groups', where as later it is discussed as four.

Reply: Thank the reviewer for the comments. In this study, the level of 8 neurotransmitters in four group (normal group, AD model group, X. sorbifolia extract and the huperzine administration group) had been detected and the aim of this paper was to evaluate the effect of .X.sorbifolia. We had rewritten the section of “introduction” to avoid confusion.

Results should not be provided in the discussions section, these should be in the results section.

Reply: Thanks a lot for the advice, we had rewritten the “Results” and “Discussion” part.

Figure 2 shows 4 NTs, the figure legend says there are 8.

Reply: We were sorry for our mistake, figure 2 showed the concentration of four differential analytes and we had corrected it in the revised paper.

The discussion is overall extremely weak and confusing and comments made are entirely speculative and don't warrant inclusion without sufficient evidence to suggest that the predictions 'could' be true. Solely measuring values does not provide the evidence in this context.

Reply: Thank you for the valuable comments, we had rewritten the “Discussion” part. In this study, we aim to evaluate the effect of Xanthoceras sorbifolia Bunge by analysis of varied neurotransmitters level in brain tissue. In this paper, the neurotransmitters level of rats, including healthy rats, AD (induced by D-gal and Aβ25-35) rats and AD rats treated with two different drugs, had been analyzed thoroughly. By targeted metabolomics analysis, simultaneous determination of 8 neurotransmitters to identify differential neurotransmitters for the early diagnosis of AD and to identify sensitive biomarkers for monitoring the effect of the drug had been developed innovatively as well as constructively. Results showed that both L-tryptophan and acetyl choline may have similar characteristics: a significant change level in accordance with AD progression and a regulated tendency according to the treatment until it becomes equal to that in normal rats. Thus, we concluded that L-tryptophan and 1 acetyl choline could serve as potential target biomarkers for monitoring medical treatments and predicting AD remission. Our results aim to demonstrated the application value of brain neurotransmitters in medical treatment of AD. Moreover, with advantages of multi-target and multi-coordinated system, Chinese medicine is being widely used in illness treatment, and positive effects have been achieved. However, problems including incomplete functional mechanisms, have affected the application of traditional Chinese medicines (TCM). So, it is important to carry out TCM to combat AD pharmacodynamic evaluation on the basis of neurotransmitters metabolism response spectrum, which will provide new information about the correlation between biological endogenous substances and TCM pharmacodynamics.

Huperzine is mentioned as a treatment to one group - at no point is it explained what huperzine is, and its preparation details are poor - 0.027mg/kg is a dose, not a stock concentration (unless it is 0.027 mg per 1 kg of solvent which I presume is not the case).

Reply: Thank you for the comments. Your suggestion is greatly appreciated. Huperzine had been found through multiple studies to be effective as a medicine for helping people with neurological conditions such as Alzheimer's disease. In this study, it had been used for positive drug and were prepared in water at the concentration of 0.027 mg ml-1.

The authors claim that only a small volume of biosample is required - but at no point do they mention how much brain tissue was used (an average or range, for example).

Reply: Thank the reviewer for the comments. In the experiment, we weighted about 0.1g brain to detecting the level of neurotransmitters. The validated method of requiring a small volume bio samples with a better sensitivity can be applied to other bio samples, such as hippocampal formation in brain, which can carry out more meaningful explorations on AD and medicine evaluation.

Thanks again for the helpful advices. All the revisions provided significant help to make the manuscript more accurate and easier to understand. We hope that the revisions we made are satisfactory and the revised version of the manuscript will be acceptable for publication in Molecules. If you have any queries, please do not hesitate to contact me.

Wish you all the best!

Yours sincerely,

Kaishun Bi, Ph.D.

Professor

Shenyang Pharmaceutical University

Reviewer 3 Report

In order to explore the potential therapeutic effect of Xanthoceras sorbifolia Bunge. against Alzheimer's disease, an UHPLC-MS/MS method had been developed and validated for simultaneous determination of 8 neurotransmitters in rat brain without complex sample pre-treatments. The completely validated method has been successfully applied to compare the 8 neurotransmitters in the rat brain from the four groups of normal, Alzheimer's disease, and the oral administration group of X. sorbifolia extract and huperzine. The results indicated the crude extract of the husks from X.sorbifolia might ameliorate the impairment of learning and memory in the Alzheimer's disease animal models with similar function of AchEI as huperzine, and the established method would be especially useful for the discovery of novel drug against Alzheimer's disease.

The major comment:

To verify the efficacy of this method to be used on discovery of novel drug against AD, the data of Morris water maze testing including escape latency and probe trial should be provided.

There are some minor points should be verified:

1.      Several abbreviations such as HPLC, UV, LC, 5-HT, HFBA, DA, MS, ESI, DP, CE, CXP, LLOQ, IS, QC, MF, etc should be full spelled in the first time.

2.      In line 55: “CE” should be full spelled as “capillary electrophoresis”.

3.      In line 91-92: “Figures, Tables and Schemes” hould be deleted.

4.      In Table 4: Group N, M, X and H should be described in legend because the Materials and Method section is followed by the result section.

5.      In Materials and Methods section: the unit such as “kg-1 day-1” or “µg µl-1” should be displayed as “kg-1 day-1” or “µg µl-1”.

6.      In Figure 1: the authors should explain why is that the retention time in the brain sample did not match to the standards, such as the retention time for NE in brain sample and in standard is 1.64 and 3.06, respectively.

7.      The methods of statics analysis should be provide in the Materials and Method section, and statics analysis be included in Figure 2.

Author Response

Dear reviewer:

Thank you so much for reviewing and giving us the chance to revise our manuscript # molecules-388362 entitled “Rapid analysis of neurotransmitters in the brain tissue of Alzheimer's disease rats before and after oral administration of Xanthoceras sorbifolia Bunge by UHPLC-ESI-MS/MS ". We greatly appreciate the efforts for handling and reviewing our manuscript as well as the valuable comments. Respective revisions that provide significant help for the manuscript have been made. Please find our response to the reviewer point by point below and the relative changes made in the revised manuscript.

The major comment:

To verify the efficacy of this method to be used on discovery of novel drug against AD, the data of Morris water maze testing including escape latency and probe trial should be provided.

Reply: Thank the reviewer for the comments. We had added the data of Morris water maze testing in Supplementary Figure2, 3 and 4.

There are some minor points should be verified:

1.      Several abbreviations such as HPLC, UV, LC, 5-HT, HFBA, DA, MS, ESI, DP, CE, CXP, LLOQ, IS, QC, MF, etc should be full spelled in the first time.

Reply: Thank you for the suggestions, these abbreviations had been replaced by the full spelled.

2.      In line 55: “CE” should be full spelled as “capillary electrophoresis”.

Reply: Thank you for pointing out our mistake, we had respelled the expressions.

3.      In line 91-92: “Figures, Tables and Schemes” should be deleted.

Reply: Thank you for pointing out our mistake, we had deleted the expressions.

4.      In Table 4: Group N, M, X and H should be described in legend because the Materials and Method section is followed by the result section.

Reply: Thank the reviewer for the comments. The legend of Table 4 had been rewritten.

5.      In Materials and Methods section: the unit such as “kg-1 day-1” or “µg µl-1” should be displayed as “kg-1 day-1” or “µg µl-1”.

Reply: Thank you for pointing out our mistake, the inaccurate expression had been corrected.

6.      In Figure 1: the authors should explain why is that the retention time in the brain sample did not match to the standards, such as the retention time for NE in brain sample and in standard is 1.64 and 3.06, respectively.

Reply: Thank you for the comments. The addition of endogenous substance in brain may increase the polarity of the solution. Thus, in brain samples, many analytes had a lower retention time than in solutions. And the eight analytes were authenticated from the comparison of the retention time of analytes in brain sample and in blank sample spiked its standards. In revised Figure 1, we displayed the typical MRM chromatograms of eight standards and internal standard, the brain samples of the AD rats and a blank sample spiked analytes and IS.

7.      The methods of statics analysis should be provide in the Materials and Method section, and statics analysis be included in Figure 2.

Reply: Thanks for the comments, the methods of statics analysis had been described in the section of “4.8 Statistical analysis” and in Figure 2.

Thanks again for the helpful advices. All the revisions provided significant help to make the manuscript more accurate and easier to understand. We hope that the revisions we made are satisfactory and the revised version of the manuscript will be acceptable for publication in Molecules. If you have any queries, please do not hesitate to contact me.

Wish you all the best! 

Reviewer 4 Report

Some points should be addressed.

1.       Abstract, make it more informative. Your conclusion is too weak here. 

2.       The Introduction is so weak. Please add more. What is the main aim of this paper?

3.       Line 146. How did you preserve the analyte? Did you try to reinject fresh? There are some enzymes present on it, and faster degradation is observed before you inject the sample.

4.       You need a thorough discussion of the results. The paper is so technical. Since this is a Molecule paper, you can add structures of Q1 and Q3. Did you optimized it? Also, discuss each group properly.

Author Response

Dear reviewer:

Thank you so much for reviewing and giving us the chance to revise our manuscript # molecules-388362 entitled “Rapid analysis of neurotransmitters in the brain tissue of Alzheimer's disease rats before and after oral administration of Xanthoceras sorbifolia Bunge by UHPLC-ESI-MS/MS ". We greatly appreciate the efforts for handling and reviewing our manuscript as well as the valuable comments. Respective revisions that provide significant help for the manuscript have been made. Please find our response to the reviewer point by point below and the relative changes made in the revised manuscript.

1.       Abstract, Abstract make it more informative. Your conclusion is too weak here.

Reply: Thanks a lot for the advice, we had rewritten the “Abstract” and “Discussion” part.

2.       The Introduction is so weak. Please add more. What is the main aim of this paper?

Reply: According to the reviewer’s advice, we had rewritten the “Introduction” part. In this study, we aim to evaluate the effect of Xanthoceras sorbifolia Bunge by analysis of varied neurotransmitters level in brain tissue. With advantages of multi-target and multi-coordinated system, Chinese herb is being widely used in various illness treatment, and positive effects have been achieved. However, problems including incomplete functional mechanisms, have affected the application of traditional Chinese medicines (TCM). So, it will be important to carry out TCM to combat AD pharmacodynamic evaluation on the basis of neurotransmitters metabolism. In this paper, the neurotransmitters level of rats, including healthy rats, AD (induced by D-gal and Aβ25-35) rats and AD rats treated with two different drugs, had been analyzed thoroughly. By targeted metabolomics analysis, simultaneous determination of 8 neurotransmitters to identify differential neurotransmitters for the early diagnosis of AD and to identify sensitive biomarkers for monitoring the effect of the drug had been developed innovatively as well as constructively. Results showed that decreased dopamine, norepinephrine and acetyl choline levels and increased tryptophan level were associated with progression of AD. Especially, both L-tryptophan and acetyl choline may have similar characteristics: a significant change level in accordance with AD progression and a regulated tendency according to the treatment until it becomes equal to that in normal rats. Our results also demonstrated the application value of brain neurotransmitters in medical treatment of AD.

3.       Line 146. How did you preserve the analyte? Did you try to reinject fresh? There are some enzymes present on it, and faster degradation is observed before you inject the sample.

Reply: In the study of method validation, we investigated the stability of the method. It had been observed that the analytes in brain were stable after being placed at room temperature within 4 h and the reconstituted extract analytes were stable at 4 °C in the auto-sampler for 4 h. Moreover, all the analytes were stable when stored at -80 °C for 1 month with and through three freeze-thaw cycles. The results of method validation indicated that the 8 analytes in brain samples were found to be stable under different storage and pretreatment conditions. Meanwhile, in the pretreatment process, the brain tissue was fleetly weighted and homogenized in ice bath to avoiding degradation. Moreover, to guaranty the accuracy of the result, we choose 10% sample for incurred sample reanalysis (ISR) to assess the reproducibility of the method and the results, and the error was within 20% between the concentration of analytes from ISR and the original analysis.

4.       You need a thorough discussion of the results. The paper is so technical. Since this is a Molecule paper, you can add structures of Q1 and Q3. Did you optimized it? Also, discuss each group properly.

Reply: Thanks a lot for the comments, we had rewritten the “Discussion” part and make a more thorough discussion on the result. In this study, the optimal conditions for precursor and product ions were determined by direct infusion of the standard solutions into the mass spectrometer and the precursor and product ions with excellent ion performance were chosen for quantitation and unequivocal identification Also, other critical parameters such as Gas1, Gas2, curtain gas, declustering potential, collision energy and cell exit potential were optimized under evaluation of the ionic sensitivity. The chemical structures and full scan of product ion spectrums from 8 analytes had been shown in Supplementary Figure 4.

Thanks again for the helpful advices. All the revisions provided significant help to make the manuscript more accurate and easier to understand. We hope that the revisions we made are satisfactory and the revised version of the manuscript will be acceptable for publication in Molecules. If you have any queries, please do not hesitate to contact me.

Wish you all the best!

Yours sincerely,

Kaishun Bi, Ph.D.

Professor

Shenyang Pharmaceutical University

Round 2

Reviewer 1 Report

The manuscript is acceptable for publication in its current form

Reviewer 3 Report

It is acceptable on this edition.

Reviewer 4 Report

Accept